# Optimizing environmental safety and cell-killing potential of oncolytic Newcastle Disease virus with modifications of the V, F and HN genes

J. Fréderique de Graaf[1], Stefan van Nieuwkoop[1], Theo Bestebroer[1], Daphne Groeneveld[1], Casper H. J. van Eijck[2], Ron A. M. Fouchier[1], Bernadette G. van den Hoogen[1]*

1 Viroscience Department, Erasmus Medical Centrum, Rotterdam, The Netherlands, 2 Department of Surgery, Erasmus Medical Centrum, Rotterdam, The Netherlands

* b.vandenhoogen@erasmusmc.nl

**Data Availability Statement:** All relevant data are within the paper and its Supporting information files.

## Abstract

Newcastle Disease Virus (NDV) is an avian RNA virus, which was shown to be effective and safe for use in oncolytic viral therapy for several tumour malignancies. The presence of a multi basic cleavage site (MBCS) in the fusion protein improved its oncolytic efficacy *in vitro* and *in vivo*. However, NDV with a MBCS can be virulent in poultry. We aimed to develop an NDV with a MBCS but with reduced virulence for poultry while remaining effective in killing human tumour cells. To this end, the open reading frame of the V protein, an avian specific type I interferon antagonist, was disrupted by introducing multiple mutations. NDV with a mutated V gene was attenuated in avian cells and chicken and duck eggs. Although this virus still killed tumour cells, the efficacy was reduced compared to the virulent NDV. Introduction of various mutations in the fusion (F) and hemagglutinin-neuraminidase (HN) genes slightly improved this efficacy. Taken together, these data demonstrated that NDV with a MBCS but with abrogation of the V protein ORF and mutations in the F and HN genes can be safe for evaluation in oncolytic viral therapy.

## Introduction

In the last few decades, the use of oncolytic viruses (OVs) as therapy in cancer patients has shown to be a promising treatment strategy with encouraging results for a variety of tumours. OVs selectively infect and damage tumours either by directly killing the cells or by promoting an anti-tumour immune response towards them. Several oncolytic viruses have been tested in clinical trials with promising results, including Newcastle Disease Virus (NDV) [1–4]. NDV is a replication competent oncolytic virus belonging to the family *Paramyxoviridae* with an avian host range under normal conditions. Several studies have shown that OV therapy using NDV resulted in promising responses in several cancer types both *in vitro* and *in vivo* and was even capable of inhibiting metastases and inducing prolonged protection to tumour reoccurrence [5–7]. The direct oncolytic effect was a result of cell lysis due to viral replication, of which the

**Funding:** JFG and DG obtained funding from the Dutch Foundation "Overleven with alvleesklierkanker"; https://www.supportcasper.nl/nl/over-support-casper/stichting/ RF and SN obtained funding from NWO-TTW grant #15414 (NWO-domein Toegepaste en Technische Wetenschappen; https://www.nwo.nl/toegepaste-en-technische-wetenschappen-ttw). The funders had no role in study design, data collection and analysis, decision to publish, or preparation of the manuscript.

**Competing interests:** The authors have declared that no competing interests exist.

efficacy is dependent on the NDV strain used [7, 8]. Similar to other OVs, a part of the therapeutic effect of oncolytic NDV is based on the induction of immune responses against the virus and the cancer cells. The secretion of pro-inflammatory cytokines, such as type I interferon (IFN), leads to counter-acting immune suppressive cells and hence enhances the anti-tumour response resulting in an indirect oncolysis [9].

NDV strains are categorized in three groups based on disease severity in chickens: non-virulent (lentogenic), intermediately virulent (mesogenic) and highly virulent (velogenic). An intracerebral pathogenicity index (ICPI) in chickens is determined as a marker for virulence [10, 11]. The cleavage site in the fusion (F) protein of NDV was shown to be a major determinant of these differences in virulence [12, 13]. Previously, we have shown that a lentogenic LaSota strain in which a multi-basic amino acid sequence at the cleavage site of the F protein was engineered (NDV F3aa), making this a mesogenic virus, had significant higher efficiency in killing tumour cells *in vitro* and in murine tumour models compared to the lentogenic strain NDV F0 [8]. Our study in non-human primates showed that NDV F3aa did not result in high pathogenicity, suggesting it is safe to use as oncolytic therapeutic in humans [14]. However, mesogenic and velogenic strains can cause outbreaks of severe disease in poultry and are defined as select agents in the USA [15]. Therefore, NDV F3aa has not been applied in clinical trials yet [3].

Several studies aimed to lower the virulence of mesogenic NDV-73T strain, while keeping the oncolytic efficacy of the virus, either by changing the cleavage site to that of a non-virulent virus, or by insertion of 198 nt at the HN-L junction [16]. Alternatively, NDV-73T was armed with immune modulatory protein genes such as granulocyte–macrophage colony-stimulating factor (GM-CSF), to increase the oncolytic effect of the virus [17].

We aimed to generate a mutant NDV F3aa with reduced replication capacity in avian cells and eggs, while maintaining killing potential in human tumour cells. Park and colleagues showed that the V protein of NDV is a determinant of host range restriction, as the V protein efficiently prevents innate host defences, such the type I IFN response as well as apoptosis, in avian cells and not human cells [18]. Multiple studies have confirmed that mutations in the open reading frame (ORF) of the V protein led to increased susceptibility to IFN responses and reduced virulence in embryonated chicken eggs [19, 20]. Nevertheless, the effect of abrogating the V protein ORF in the viral genome on virus induced tumour cell killing and environmental safety is not known yet. In this study, we generated NDV F3aa mutants in which expression of the V protein was abolished by introducing mutations in the stutter site and throughout the V protein ORF (NDV F3aa-$_{STOP}$V). The mutations introduced in the V protein ORF affected all of the third positions of the codons of the essential Phosphoprotein (P) gene which did not result in any amino acid substitutions in the P protein but resulted in 15 stop codons in the V protein ORF. We investigated the effect of these modifications on avian-specific virulence. To improve the replication efficiency and oncolytic efficacy of these NDV F3aa mutants, additional mutations were introduced in the MBCS (F-117-S) and the intracellular domain (ICD) of the F protein (Y-524-A), which were suggested to improve F protein expression [21, 22]. In addition, the translation initiation site (TIS) of the F and HN genes was mutated to an optimized mammalian Kozak consensus sequence, aimed to obtain improvement of protein expression and hence viral replication [23]. These mutant viruses were evaluated for replication efficiency and cell-killing potential in human pancreatic adenocarcinoma cells (HPACs).

## Results

### Generation of NDV F3aa lacking V protein expression

The ORFs of the V and W proteins overlap with the ORF of the essential phospho (P) protein. The V protein ORF is only transcribed after a frameshift occurs as the result of stuttering by

the viral polymerase protein [24]. To produce viruses that lack V protein expression, the stutter site was mutated at one or two nucleotide positions. In addition, every third nucleotide of the P protein codons was mutated, but only if this mutation did not result in a change of amino acid sequence of the P protein. In total 95 mutations were introduced, which resulted in the introduction of 15 stop codons in the V protein ORF, yielding the viruses NDV F3aa-$_{STOP}$V1 and NDV F3aa-$_{STOP}$V2, respectively (Fig 1A and 1B). The amino acid sequence of the P protein remained unchanged, because the mutations introduced into the ORF of the V protein only affected the third positions of the codons of the P protein gene. However, the modifications of the V protein ORF resulted in 40 amino acid substitutions in the W protein ORF including the deletion of the stop codon. Recombinant viruses were rescued in BSRT-7 cells and propagated further in Vero cells. The V protein gene of recombinant virus stocks from passage 2 were sequenced, which revealed that the viruses maintained the intended V protein sequence as indicated in Fig 1B.

## Replication kinetics of NDV F3aa lacking V protein expression *in vitro*

To assess possible attenuation of the mutant viruses, replication kinetics of the $_{STOP}$V viruses were evaluated in two different avian cell lines and in human adenocarcinoma cells known as a IFN responsive type II pulmonary epithelial cell (A549 cells). In chicken fibroblasts (DF-1 cells), the NDV F3aa-$_{STOP}$V viruses were slightly attenuated (Fig 1C), while no replication was observed in quail fibroblasts (QT6 cells) (Fig 1D). Compared to the F3aa virus, the F3aa-$_{STOP}$V viruses were severely attenuated in A549 cells (Fig 1E). Thus, the abolishment of V protein expression from NDV F3aa resulted in substantially attenuated virus replication in avian and human cells.

## Increased interferon sensitivity of NDV F3aa lacking V protein expression

To evaluate the IFN sensitivity of the F3aa-$_{STOP}$V viruses *in vitro*, chicken fibroblasts were pretreated with chicken IFN-β before virus inoculation. In these IFN-β pretreated cells, NDV F3aa replication was not affected (Fig 2A). In contrast, replication of both NDV F3aa-$_{STOP}$V viruses was reduced in IFN treated cells compared to untreated cells, of which the replication of NDV F3aa-$_{STOP}$V2 was significantly attenuated. These data indicate that abrogation of V protein expression increased the sensitivity of the viruses to IFN.

To examine whether the loss of V protein expression also led to increased IFN responses, IFN-β mRNA expression levels upon inoculation of chicken DF-1 cells and human A549 cells were determined. Inoculation of DF-1 cells with NDV-F3aa did not result in elevated IFN-β mRNA levels, whereas inoculation with both NDV F3aa-$_{STOP}$V viruses resulted in increased IFN-β mRNA expression levels compared to mock inoculation (Fig 2B). Inoculation of DF-1 cells with NDV F3aa-$_{STOP}$V2 resulted in lower IFN IFN-β mRNA expression levels than inoculation with NDV F3aa-$_{STOP}$V1, which is probably due to the higher levels of attenuation of NDV F3aa-$_{STOP}$V2 in cells pretreated or producing INF, as shown in Fig 2A. These results confirm that the V protein acts as species specific IFN antagonist in avian cells [18]. In human A549 cells, all viruses induced increased IFN-β mRNA levels but differences were not observed between the viruses with and without an intact V ORF (Fig 2C).

To confirm that the NDV F3aa-$_{STOP}$V viruses were also attenuated in eggs due to increased IFN sensitivity, virus replication was assessed in embryonated chicken and duck eggs of different ages. Embryonated chicken eggs older than 10 days and duck eggs older than 13 days were described to produce IFN, in contrast to younger eggs [25]. Inoculation of chicken and duck eggs of all ages with NDV F3aa resulted in higher virus titers than inoculation with NDV F0 or F3aa-$_{STOP}$V viruses (Fig 2D and 2E). In 6-day old chicken eggs, which have been reported to

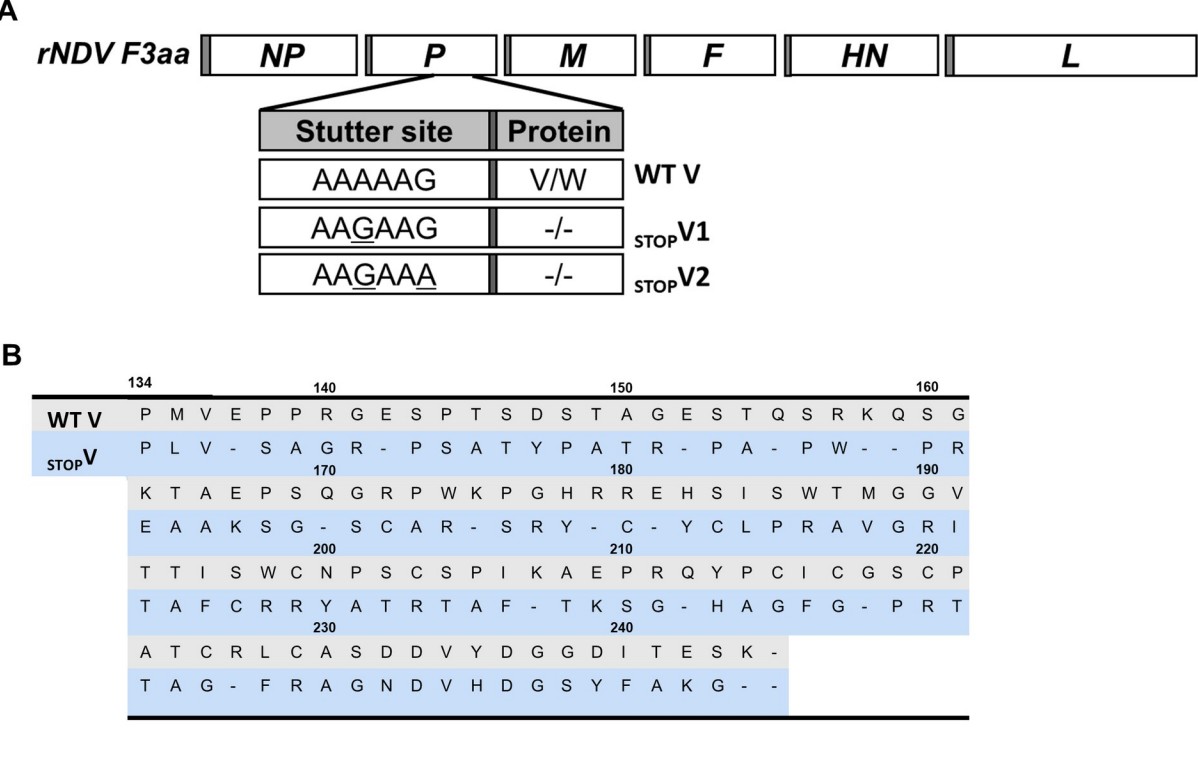

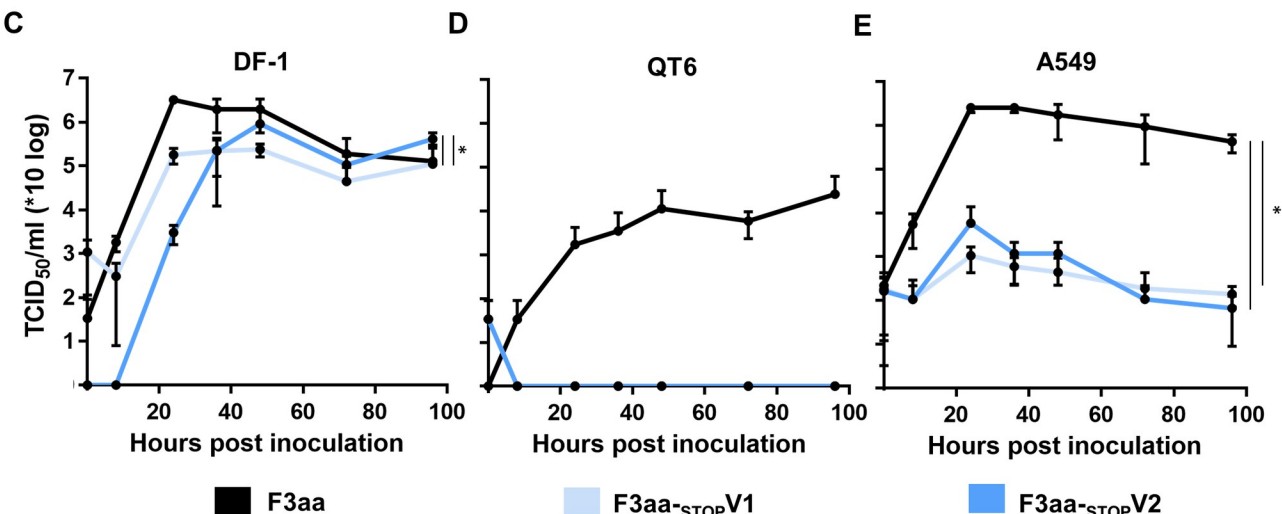

**Fig 1. Characterization of NDV F3aa-STOPV viruses *in vitro*.** (**A**) Schematic representation of the sequence of the stutter site of NDV F3aa and the NDV F3aa-STOPV mutants. (**B**) Amino acid sequence of V protein ORFs of NDV F3aa and F3aa-STOPV. (**C-E**) Cells were inoculated in triplo in (**C**) DF-1 at an MOI of 0.005 and at an MOI of 0.05 in (**D**) QT6 and (**E**) A549 cells. Supernatant samples were collected at the indicated time points and titrated in Vero cells (N = 2). Means and standard deviations of triplicates of representative experiments are plotted. The area under the curve (AUC) was used for statistical analysis. * = p <0.05, ** = p<0.01, *** = p<0.001 (one-way ANOVA + paired t-test).

lack IFN production, higher virus titers of NDV F3aa-STOPV mutants were obtained than in older eggs. Upon inoculation of 14-day old chicken eggs which have been reported to produce IFN with NDV F3aa-STOPV2, no virus titers were obtained at all, underpinning the higher degree of attenuation of NDV F3aa-STOPV2 compared to that of NDV F3aa-STOPV1. Surprisingly, inoculation with NDV F0 led to higher titers in older eggs than in younger eggs.

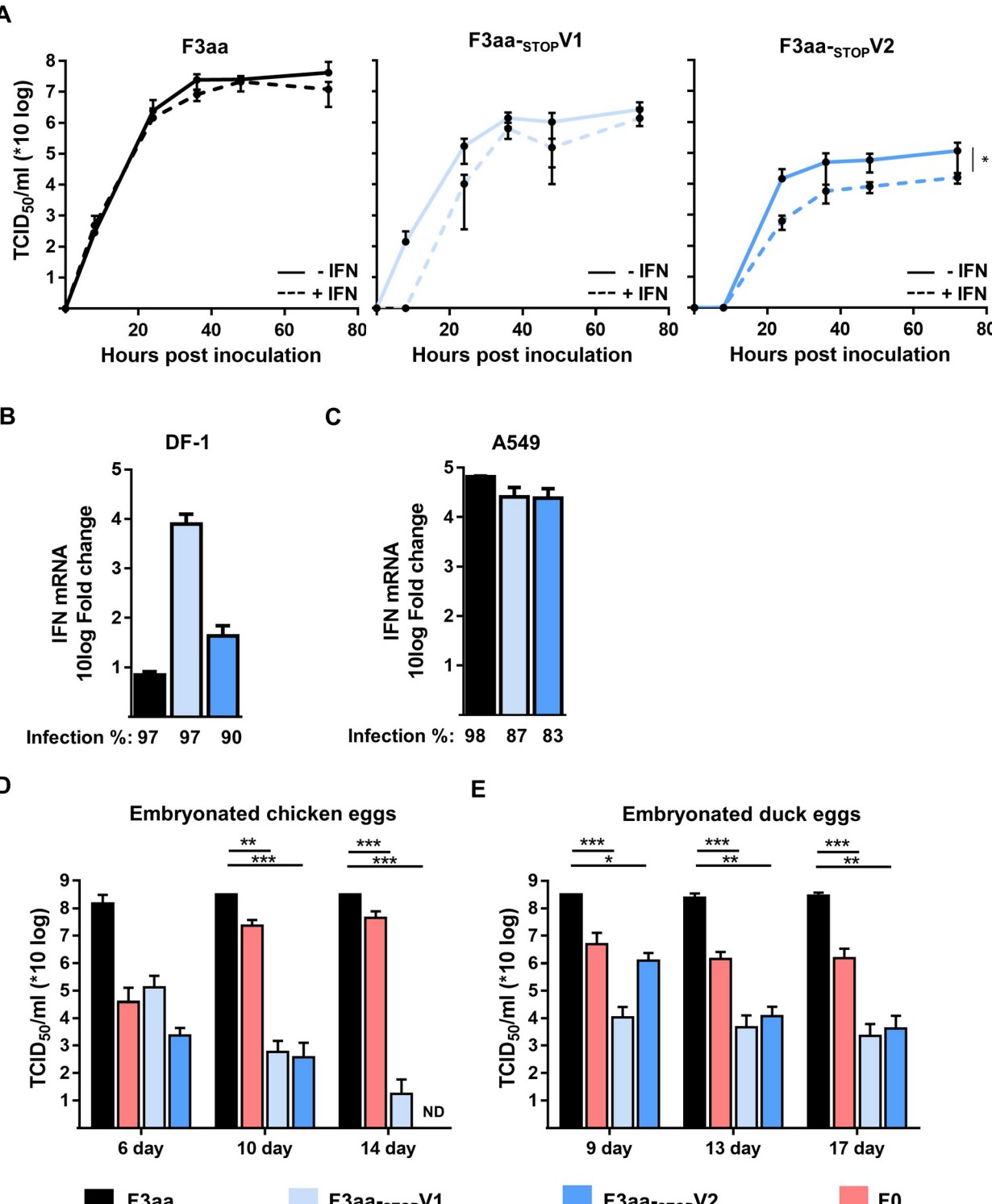

**Fig 2. Characterization of NDV F3aa-STOPV viruses *in ovo*.** (**A**) DF-1 cells were mock-treated (solid line) or pretreated with 0.4 μg/ml chicken IFN-β for 24 hrs (dotted line) and inoculated in triplo at an MOI of 0.005. Samples were taken at the indicated time points and titrated. The experiment was conducted two times. Means and standard deviations of triplicates of a representative experiment are plotted. The AUC was used for statistical analysis. * = p <0.05, ** = p<0.01, *** = p<0.001 (one-way ANOVA + paired t-test). (**B**) DF-1 or (**C**) A549 cells were inoculated with mock or at an MOI of 0.05 and harvested after 24h. The percentage of infected cells was determined by flow cytometry. Results are represented as fold change of IFN mRNA transcription in virus-infected cells versus mock. (**D**) Chicken and (**E**) duck eggs of different ages were inoculated and the

allantoic fluid was harvested after 48 hours. The amount of infectious virus particles was determined by titration, with a maximal cut off of $8.3E^8$ TCID50/ml. * = p <0.05, ** = p<0.01, *** = p<0.001 (one-way ANOVA + Dunn's multiple comparison test). ND: not detected.

Inoculation of duck eggs with NDV F3aa-$_{STOP}$V mutants resulted in similar virus titers in both young and older eggs, although inoculation of younger eggs with NDV F3aa-$_{STOP}$V2, resulted in slightly higher titers than inoculation of older eggs. These data indicate that abolishment of V protein expression in NDV F3aa-$_{STOP}$V resulted in an attenuated replication in both chicken and duck eggs, which in chicken eggs seems to be in part related to the age of the eggs, probably as a consequence of increased IFN sensitivity of the viruses.

## Effect of NDV F3aa lacking V protein expression on tumour cell killing

The virus induced killing of tumour cells was assessed by determining the cell viability upon inoculation of ten HPAC cell lines. In general, inoculation of HPACs with NDV F3aa resulted in more cell death than inoculation with the other viruses (Fig 3A) and inoculation with NDV F3aa-$_{STOP}$V viruses induced a similar level of cell killing as inoculation with NDV F0 in most HPACS, but not in Su.86.86 and CFPAC. The virus induced cell killing did not always correspond with the peak virus titers obtained at 48 hours after inoculation (Fig 3B). For instance, upon inoculation of CFPAC cells, NDV F3aa-$_{STOP}$V induced less cell killing than NDV F0, while peak virus titers were similar for these viruses. Similarly, although similar virus titers were obtained upon inoculation of CPFAC and HPAFII cells with F3aa-$_{STOP}$V, more cell killing was observed in CFPAC than in HPAFII cells. In two HPACS that were selected because they were most sensitive and least sensitive to NDV (Fig 3A), Su.86.86 and AsPC-1 respectively, replication of F3aa-$_{STOP}$V viruses was clearly attenuated compared to NDV F3aa (Fig 3C and 3D). Although the replication of NDV F3aa-$_{STOP}$V viruses was attenuated compared to NDV F3aa, the observed killing of tumour cells, being similar to that induced by NDV F0 in most cell lines, indicate that the viruses still have oncolytic potential.

## Genomic stability of NDV F3aa-$_{STOP}$V viruses

During passaging of the recombinant viruses in mammalian and chicken cells seven times, all mutations introduced in both the stutter site and the open reading of the V protein were found to be genetically stable. However, in all virus stocks passaged in Vero cells, a phenylalanine to serine substitution occurred at position 117 in the F protein (F-117-S), located at the cleavage site of the protein. This substitution was not observed in any of the viruses passaged in chicken DF-1 cells.

To evaluate the effect of the F-117-S substitution on virus production and cell death in HPACs, the substitution was introduced in the backbone of the NDV F3aa and NDV F3aa-$_{STOP}$V2 viruses, resulting in the viruses NDV F3aa-S and NDV F3aa-S-$_{STOP}$V2 (Fig 4A). Given that NDV F3aa-$_{STOP}$V1 and NDV F3aa-$_{STOP}$V2 induced similar responses in HPACs and embryonated eggs (Figs 1–3), we continued only with the virus with 2 mutations in the stutter-site as this virus has a smaller hypothetical chance of reversion. The F-117-S substitution did not result in substantial differences in virus induced cell killing for the NDV F3aa-S and NDV F3aa-S-$_{STOP}$V2 viruses upon inoculation of four different HPACs (Fig 4B). However, increased cytotoxicity was observed upon inoculation of Vero cells with viruses with F-117-S for both NDV F3aa-S as NDV F3aa-S-$_{STOP}$V2 (Fig 4C). In addition, slightly higher replication kinetics in Vero cells were observed for the NDV F3aa-S virus than for NDV F3aa virus and a larger increase in replication was seen for NDV F3aa-S-$_{STOP}$V2 as compared to NDV F3aa-$_{STOP}$V2 (Fig 4D). The improved replication of NDV F3aa-S-$_{STOP}$V in Vero cells could be

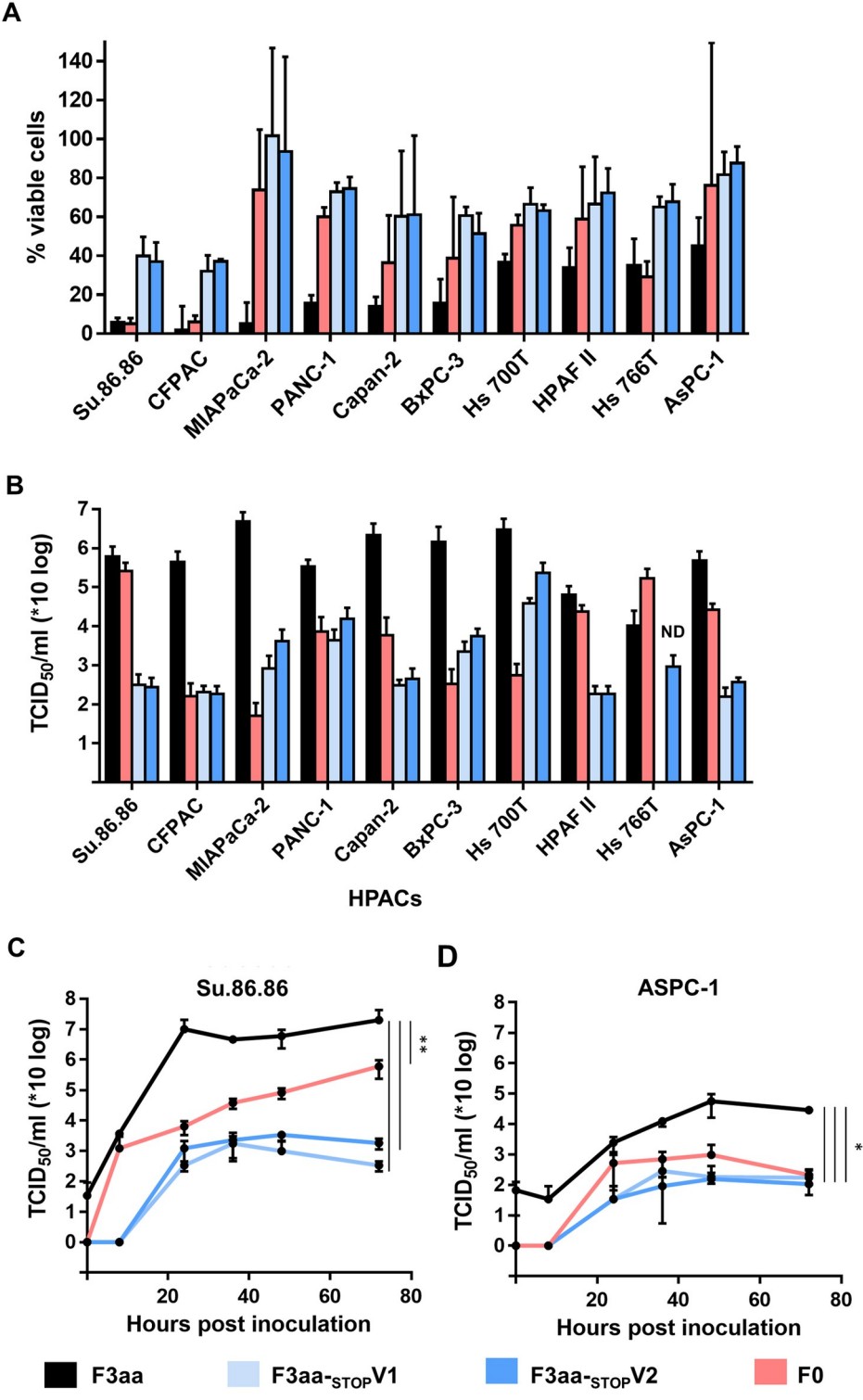

**Fig 3. Cell death of human cancer cell lines upon inoculation with NDV F3aa-STOPV viruses. (A)** Indicated HPACs were inoculated at an MOI of 10 in triplo with the indicated viruses. Cell viability was determined by an LDH cytotoxicity assay 120 hours after inoculation. Results are represented as percentage viable cells compared to mock, which were considered as 100% viable. Experiments were conducted two times. Means and standard deviations of triplicates of representative experiments are plotted. (**B**) Indicated HPACs were inoculated in duplo at an MOI of 0.1. After 48 hours, supernatant samples were collected and titrated in Vero cells. The experiment was conducted two

times. Means and standard deviations of triplicates of a representative experiment are plotted. (**C**) Su.86.86 and (**D**) AsPC-1 cells were inoculated at an MOI of 0.05 in triplo. Supernatant samples were collected at the indicated time points and titrated in Vero cells. The experiment was conducted two times. Means and standard deviations of triplicates of a representative experiment are plotted. The AUC was used for statistical analysis. * = p <0.05, ** = p<0.01, *** = p<0.001 (one-way ANOVA + paired t-test). ND: not detected.

beneficial for obtaining high virus titers during the production of oncolytic viruses for cancer therapy.

To investigate whether the improved replication in Vero cells was due to improved fusion activity as a consequence of the substitution in the F protein cleavage site, fusion assays were performed. In these assays, cells were inoculated with virus and fixed 16 hours later followed by Giemsa staining. Subsequently, the number of nuclei per foci were counted to determine the fusion activity of each virus. NDV F3aa-S and NDV F3aa-S-<sub>STOP</sub>V2 viruses showed significantly increased fusion activity in Vero cells compared to viruses without F-117-S and compared to NDV F0 (Fig 4E). The beneficial effect of F-117-S on the fusion activity of NDV F3aa-S and NDV F3aa-S-<sub>STOP</sub>V2 viruses was not observed in DF-1 cells (Fig 4F). These data indicate that the F-117-S mutation probably provides a specific adaptation towards Vero cells.

## Reduced virulence of NDV F3aa-(S)-<sub>STOP</sub>V2 viruses

To further assess the virulence of NDV F3aa-<sub>STOP</sub>V2 and NDV F3aa-S-<sub>STOP</sub>V2, the mean death time (MDT) of embryonated chicken and duck eggs upon inoculation with these viruses was determined as compared to viruses with an intact V protein ORF (Table 1). A MDT of chicken embryos upon NDV inoculation greater than 90 hours has been shown to be typical for nonvirulent strains that have an ICPI value smaller than 0.7 [26]. Upon inoculation with NDV F3aa and NDV F3aa-S, chicken eggs reached an MDT below 60 hours and duck eggs below 100 hours. Inoculation with NDV F0 resulted in an MDT just above 100 hours for eggs of both origins. No MDT could be determined for both chicken and duck eggs inoculated with NDV F3aa-<sub>STOP</sub>V2 and NDV F3aa-S-<sub>STOP</sub>V2, because all embryos survived. These data indicate that NDV F3aa-<sub>STOP</sub>V2 and NDV F3aa-S-<sub>STOP</sub>V2 were nonvirulent for poultry embryos, in contrast to NDV F3aa and NDV F3aa-S.

## Mutations in the F and/or HN gene improved replication of NDV F3aa-S-<sub>STOP</sub>V2

The attenuation of NDV F3aa-(S)-<sub>STOP</sub>V2 mutants in avian cells and eggs was accompanied by reduced virus replication and virus-induced cell death in most HPACs. To improve virus replication and virus-induced cell killing, we introduced a number of mutations in the F and HN genes of the NDV F3aa-S-<sub>STOP</sub>V2 virus (Fig 5A). First of all, a substitution (Y-524-A) was introduced in the intracellular domain (ICD) of the F protein. This substitution was previously shown to cause increased virus replication, possibly due to improved stability of the F protein [21]. Introduction of this substitution in the genome of NDV F3aa-S-<sub>STOP</sub>V2, yielded the virus NDV F3aa-S-<sub>STOP</sub>V2-ICD. In addition, the translation initiation sites (TIS) of the F and HN proteins were changed to a mammalian optimal TIS sequence, the Kozak consensus sequence, with the aim to increase the translation and expression of both proteins [23]. This yielded NDV F3aa-S-<sub>STOP</sub>V2-TIS (Fig 5A). The NDV F3aa-S-<sub>STOP</sub>V2-ICD and NDV F3aa-S-<sub>STOP</sub>V2-TIS viruses were genetically stable until at least passage 4 in Vero cells.

In Vero cells, the replication of NDV F3aa-S-<sub>STOP</sub>V2-TIS and NDV F3aa-S-<sub>STOP</sub>V2-ICD was attenuated compared to NDV F3aa-S-<sub>STOP</sub>V2 but the viruses reached similar end titers (Fig 5B). In contrast, in the HPAC cell line Su.86.86, replication of NDV F3aa-S-<sub>STOP</sub>V2-ICD

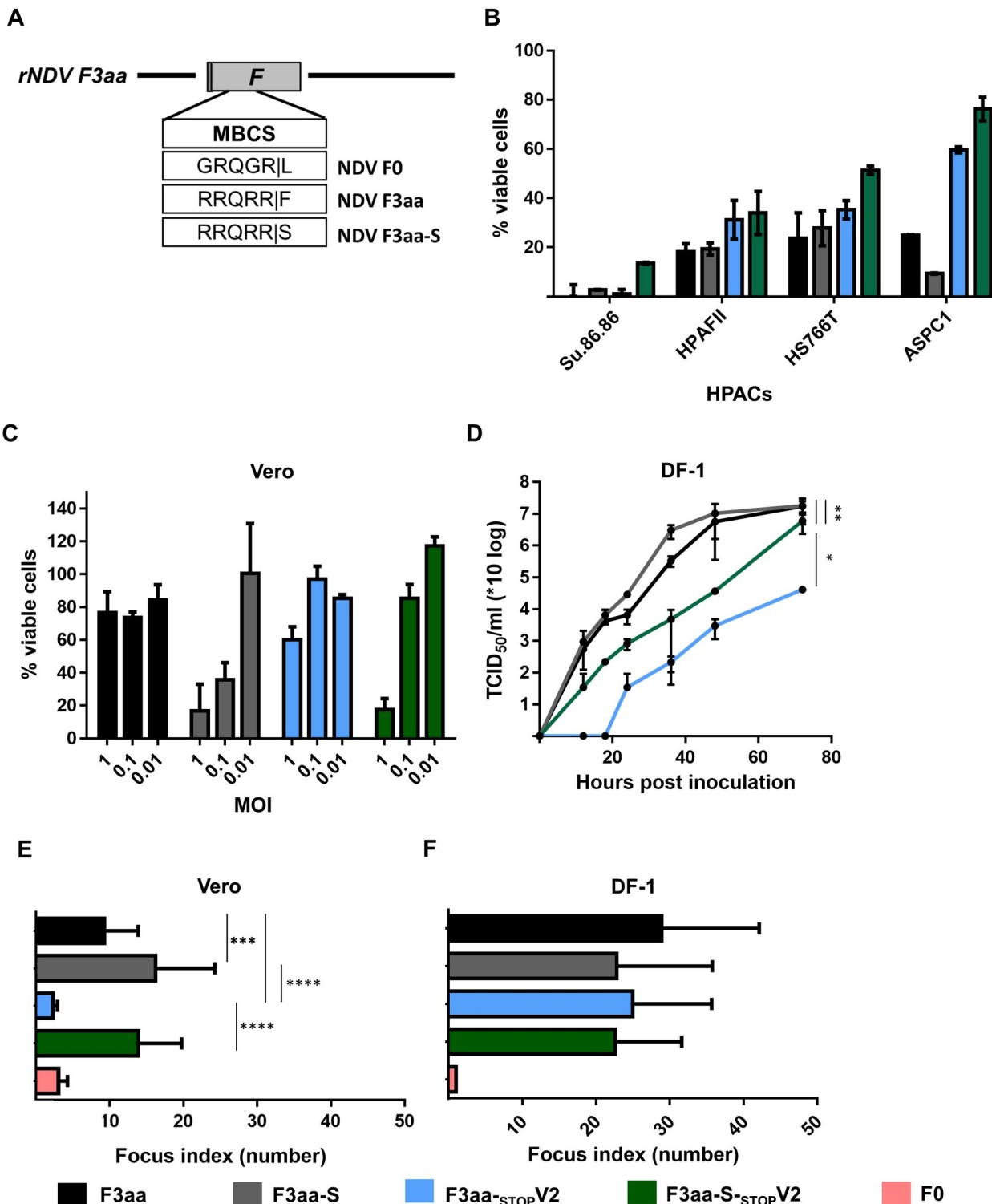

**Fig 4. Characterization of F3aa-S mutant viruses.** (**A**) Schematic representation of the amino acid sequence of the cleavage site in the F protein of recombinant viruses. (**B**) Indicated HPACs were inoculated at an MOI of 10 or (**C**) Vero cells at indicated MOIs (Vero cells) in triplo. The percentage viable cells was determined by an LDH cytotoxicity assay 120 hours after inoculation. Results are represented as percentage viable cells compared to mock, which were considered 100% viable. (**D**) Vero cells were inoculated at an MOI of 0.05 in triplo. Supernatant samples were collected at the indicated time points and titrated. The AUC was used for statistical analysis. * = p <0.05, ** = p<0.01, *** = p<0.001 (one-way ANOVA + paired t-test). (**E**) Vero and (**F**) DF-1 cells were inoculated and fixed 16 hours later and stained. The focus index was determined by

counting the number of nuclei per foci for N = 30 foci. Experiments were conducted twice. Means and standard deviations of triplicates of a representative experiment are plotted. *** = p<0.001, **** = p<0.0001 (one-way ANOVA + unpaired t test). NDV F0, taken along as control.

was slightly higher than that of NDV F3aa-S-$_{STOP}$V2 and F3aa-S-$_{STOP}$V2-TIS (Fig 5C). In ASPC-1 cells, replication of both NDV F3aa-S-$_{STOP}$V2-TIS and NDV F3aa-S-$_{STOP}$V2-ICD was higher than that of NDV F3aa-S-$_{STOP}$V2 (Fig 5D). The higher replication of NDV F3aa-S-$_{STOP}$V2-TIS and NDV F3aa-S-$_{STOP}$V2-ICD in ASPC-1 cells as compared to NDV F3aa-S-$_{STOP}$V2 did not result in increased cell death of these HPACs. Only in 2 of the HPACs (Su.86.86 and HPAFII), inoculation with NDV F3aa-S-$_{STOP}$V2-TIS and NDV F3aa-S-$_{STOP}$V2-ICD resulted in more cell death than inoculation with NDV F3aa-S-$_{STOP}$V2. Inoculation of CFPAC and Capan-2 cells resulted in more cell death after inoculation with NDV F3aa-S-$_{STOP}$V2-TIS than upon inoculation with NDV F3aa-S-$_{STOP}$V2 or NDV F3aa-S-$_{STOP}$V2-ICD (Fig 5E). In the other HPACS virus induced cell killing was similar for the three viruses. Thus, the introduced mutations in the F and HN genes of the NDV F3aa-S-$_{STOP}$V2 virus improved virus replication and cell killing of NDV F3aa-S-$_{STOP}$V2 somewhat, but not consistently across all cell lines tested.

Examination of the virulence in embryonated chicken eggs demonstrated that NDV F3aa-S-$_{STOP}$V2-TIS and NDV F3aa-S-$_{STOP}$V2-ICD did not kill chicken embryos (MDT >160 hours, Table 1). These observations were similar to those seen for the parental NDV F3aa-S-$_{STOP}$V2 virus, indicative of the nonvirulent phenotype of NDV F3aa-S-$_{STOP}$V2-ICD and NDV F3aa-S-$_{STOP}$V2-TIS viruses. The virulence in duck eggs was not investigated for NDV F3aa-S-$_{STOP}$V2-TIS and NDV F3aa-S-$_{STOP}$V2-ICD.

## Discussion

Oncolytic viro immunotherapy, based on OVs, is rapidly gaining interest in the field of immunotherapy against cancer. Due to significant virulence associated with the use of some of the human pathogens, animal viruses have been explored as an alternative, with NDV being one of them. NDV does not cause disease in humans, the virus does not interact with the host cell DNA, cellular genome integration is impossible, and the virus can efficiently and selectively replicate in tumor cells. In addition, NDV has been shown to be very safe upon high dose administration in humans, with no serious adverse events noted in early clinical trials. The results of early trials with NDV have been relatively disappointing, but with the advent of recombinant DNA techniques it has become possible to genetically engineer NDV and interest in the use of recombinant NDV (rNDV) as an oncolytic virus has revived over the last decade.

**Table 1. Mean death time of chicken and duck eggs upon inoculation with NDV and engineered mutants thereof.**

| Virus | Chicken egg MDT (h)* | Duck egg MDT (h) |
|---|---|---|
| NDV F0 | 107 | 32 |
| NDV F3aa | 52 | 87 |
| NDV F3aa-S | 58 | 96 |
| NDV F3aa—$_{STOP}$V2 | >160 | >184 |
| NDV F3aa-S—$_{STOP}$V2 | >160 | >184 |
| NDV F3aa-S—$_{STOP}$V2-ICD | >160 | N.D. |
| NDV F3aa-S—$_{STOP}$V2-TIS | >160 | N.D. |

* A MDT of chicken eggs >90h was shown previously to be typical for nonvirulent NDV strains [19].
N.D.—Not done.

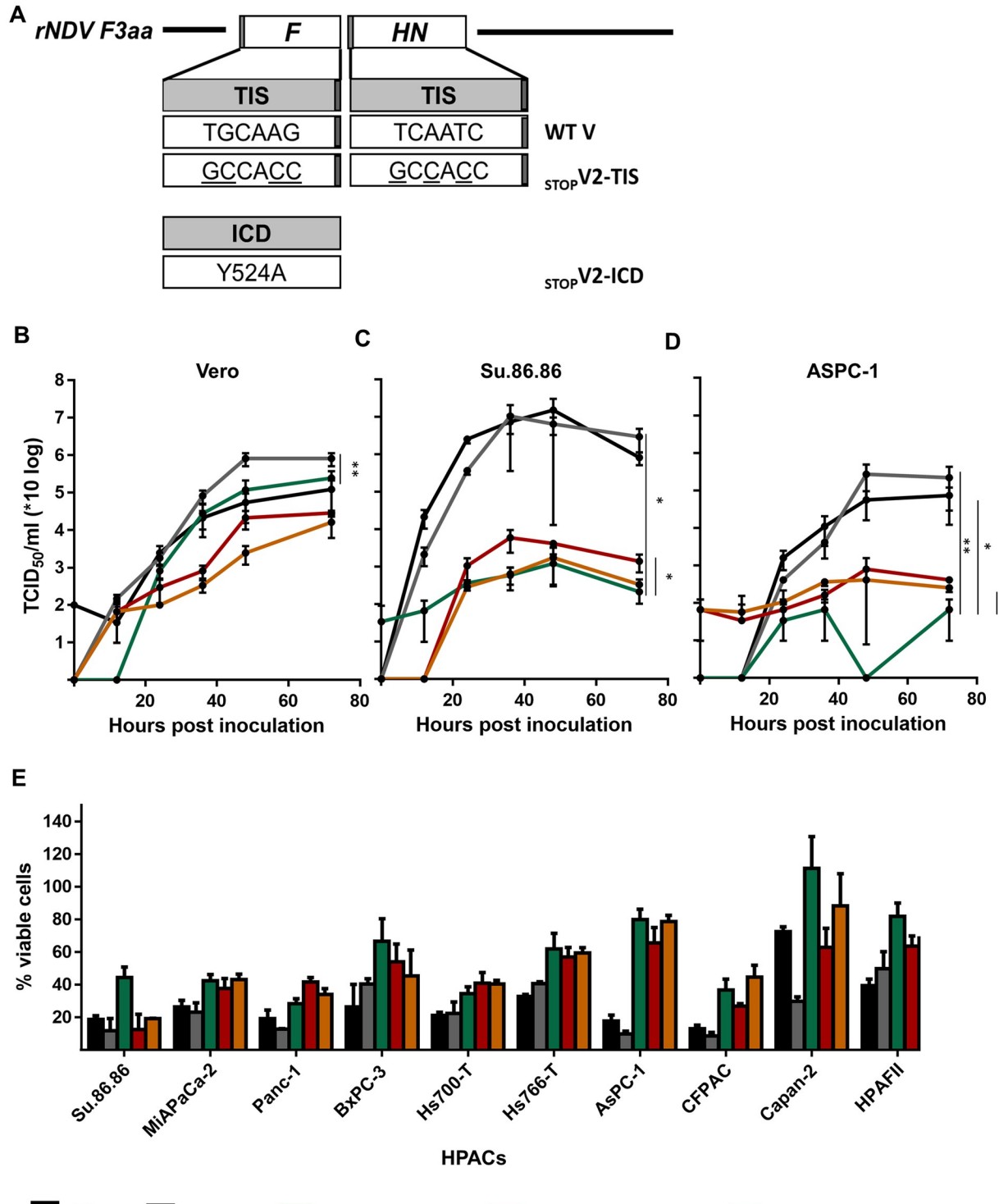

**Fig 5. Characterization of NDV F3aa-(S)-$_{STOP}$V mutants.** (**A**) Schematic representation of the nucleotide sequence of the TIS and amino acid substitution of the ICD of the F and HN protein. (**B**) Vero cells, (**C**) Su.86.86 and (**D**) AsPC-1 were inoculated at an MOI of 0.05 in triplo. Supernatant samples were collected at the indicated time points and titrated. Means and standard deviations of triplicates of a representative experiment are plotted. The AUC was used for statistical analysis. $^* = p <0.05$, $^{**} = p<0.01$, $^{***} = p<0.001$ (one-way ANOVA + paired t-test as compared to NDV F3aa-S- $_{STOP}$V2). (**E**) The indicated HPACs were inoculated at an MOI of 10 in triplo. The percentage viable cells was determined by an LDH cytotoxicity assay. Results are represented as percentage viable cells compared to mock, which were considered 100% viable. The experiment was conducted two times. Means and standard deviations of triplicates of a representative experiment are plotted.

One of the advantages of NDV is that genetic engineering of the genome is relatively easily, and the genome is forgiving for the acceptance of foreign genes. However, the limitation of using NDV is their potential to become virulent avian species.

Although, several studies have demonstrated the safety and efficacy of lentogenic NDV F0 in clinical trials [1, 2], the use of mesogenic NDV F3aa resulted in higher oncolytic efficacy *in vitro* and *in vivo*. [8] the use of such a strain in oncolytic viral therapy could theoretically pose a risk for poultry due to the possible shedding of virus by treated patients [3, 4]. To avert any risks of outbreaks with NDV F3aa, we aimed to develop a virus that has oncolytic potential but does not cause disease in avian species. To this end, we abrogated the expression of the V protein, a specific avian IFN antagonist, by mutation of the stutter site and introduction of several stop codons in the ORF of the V protein, yielding viruses NDV F3aa-$_{STOP}$V1 and NDV F3aa-$_{STOP}$V2, and evaluated the genetic stability and their virulence *in vitro* and *in ovo*.

Analyses of the genetic stability upon passaging the mutant viruses in chicken and mammalian cells revealed that the mutations introduced in the V protein ORF were genetically stable. However, during passaging in Vero cells, a F-117-S substitution in the MBCS of the F protein was observed. Introduction of this substitution in NDV F3aa and NDV F3aa-$_{STOP}$V2 resulted in increased virus replication, fusion activity and cell death in Vero cells, but not in increased cell death in most HPAC cell lines. Our fusion experiment results are in agreement with previously reported data on the characterization of NDV with the F-117-S substitution in *in vitro* experiments showing that F-117-S resulted in higher fusion activity in HT1080 and Hela cells. However, in that study, the NDV 73 T strain with the F-117-S substitution had an intracerebral pathogenicity index (ICPI) of 1.05 in chickens, substantially higher than the ICPI of 0.13 for the wild type virus [22]. In contrast, in our study the LaSota variant of NDV F3aa-S had a slightly delayed MDT of chicken and duck embryos as compared to NDV F3aa, indicative for a slightly lower virulence. These observed differences in virulence might be due to the use of different methods for determining the virulence of NDV, e.g. MDT vs ICPI or due to differences between the NDV LaSota and NDV 73 T strains.

The NDV F3aa-$_{STOP}$V viruses displayed attenuated replication in avian and human cells. This attenuation was more pronounced in IFN-pretreated chicken cells. In addition, inoculation of chicken cells with the NDV F3aa-$_{STOP}$V viruses induced higher IFN mRNA levels than the virus with an intact V protein, but this difference was not observed in human cells. Multiple studies have reported comparable results on viruses with a single mutation in the stutter site or introduction of a single stop codon in the V protein ORF, which led to increased susceptibility to IFN [18, 24, 27]. Our results are in agreement with these studies and confirm that the disruption of the V protein ORF attenuates NDV, likely due to increased IFN sensitivity *in vitro*. Similar results were obtained upon inoculation of chicken eggs, in which reduced virus titers were observed for NDV without a V protein [24].

Our study demonstrated that NDV F3aa-$_{STOP}$V viruses were attenuated in both chicken and duck eggs. Virus titers for NDV F3aa-$_{STOP}$V were slightly higher in young eggs as compared to older eggs, which is in agreement with the fact that older eggs produce IFN, suggesting that the attenuation *in ovo* is also based on increased IFN sensitivity. Based on the analyses of the MDT of eggs, both the NDV F3aa-$_{STOP}$V and F3aa-S-$_{STOP}$V should be considered as nonvirulent (lentogenic) viruses, compared to the mesogenic NDV F3aa and NDV F3aa-S viruses. Several studies have reported that eggs inoculated with viruses containing only a single stop codon in the V protein ORF had a similar or slightly higher MDT than eggs inoculated with the wild type virus [19, 28]. This difference in MDT in our study might be the result of additional amino acid substitutions in the W protein of our NDV F3aa-(S)-$_{STOP}$V viruses. The function of the W protein has not been studied extensively, but it has been shown that the protein is expressed and does not function as an IFN antagonist [19, 29]. The attenuation of the

NDV F3aa-STOPV viruses for poultry cells and eggs indicate that these viruses can be safely evaluated in oncolytic viral therapies.

However, it is not known yet whether NDV F3aa-STOPV is able to cause disease in adult chickens. This is important information for the unlikely event that treated patients spread virus to birds. In addition, at present, all NDV strains which possess an MBCS in their fusion protein, or have an ICPI above 0.7, are considered to be select agents. Like the virulence in adult chickens, the ICPI has not been determined yet for the NDV F3aa-STOPV viruses. Based on the MDT, it is expected that these viruses have an ICPI lower than 0.7, classifying the viruses as lentogenic. However, the presence of an MBCS in the fusion protein, would still categorize the virus as virulent. Perhaps, an attenuated strain of a virus with an MBCS could be approved for viro-immune-therapy studies by the regulatory instances when virulence studies indicate the attenuated nature of these viruses.

Although the NDV F3aa-(S)-STOPV viruses were slightly attenuated in mammalian cells, these viruses were still able to replicate in the HPACs and induced cell death.

Mutating the ICD of the F protein or the TIS of the F and HN proteins slightly improved virus replication of the NDV F3aa-S-STOPV2 virus in HPACs, but not to similar levels as those of NDV F3aa. The improved replication of the ICD mutant in AsPC-1 is in line with previously reported studies [21]. In addition, the introduction of the TIS mutation in NDV F3aa-S-STOPV2 did increase virus induced cell death in some HPACs. Additional adaptations of these viruses, such as the incorporation of immune modulatory genes to increase anti-tumour responses, may further improve the therapeutic efficacy of our NDV mutants [30].

Taken together, our data indicate that NDV F3aa(-S)-STOPV viruses are attenuated for avian species but are perhaps over-attenuated to kill tumour cells as efficiently as NDV F3aa. This attenuation was not completely overcome with the introduced mutations (TIS, ICD) investigated in this study. Further adaptations and *in vivo* studies are necessary to confirm the safety and effectivity of NDV F3aa-STOPV viruses in oncolytic viral therapies in the future.

## Methods

### Cell lines

The avian cell lines DF-1 and QT6 were both obtained from the American Type Culture Collection. DF-1 cells were cultured at 39°C in Dulbecco's Modified Eagle's Medium (DMEM, Lonza, The Netherlands) supplemented with 100 U ml$^{-1}$ penicillin, 100 U ml$^{-1}$ streptomycin, 2mM L-glutamine (PSG) and 10% Hyclone Characterized Fetal Bovine serum (FBS HC, Thermo Fischer Scientific, The Netherlands). QT6 cells were cultured at 37°C in Medium 119 (Lonza) supplemented with PSG, 5% FBS HC, 1% chicken serum (Sigma-Aldrich, Germany) and 5% tryptose phosphate broth (MP Biomedicals, Belgium). BSR-T7 (kind gift of K. Conzelmann) were cultured in DMEM supplemented with PSG and 10% FBS FC at 37°C. A549 were cultured in HAM's F-12 (GIBCO, Life Technologies, The Netherlands). Vero cells and human pancreatic adenocarcinoma cell lines were cultured as previously described [31]. In case of virus infection experiments, 2% FBS was used in all media (infection media).

### Cloning of recombinant viruses

The reversed genetics system for the NDV LaSota strain has been described before and was kindly provided by Ben Peeters from the Central Veterinary Institute of Wageningen, The Netherlands [32]. To introduce mutations in the V ORF a subclone was generated, using the unique digestion sites Sac-II and Not-I. The V protein with introduced mutations coding for stop codons (NDV F3aa-STOPV1 or STOPV2, Fig 1B) was ordered at Integrated DNA Technologies (IDT, Iowa) and inserted via PCR-cloning using the V protein specific insertion primers

**Table 2. Primers used for cloning.**

| Construct | Primer | Sequence 5'- 3' |
|---|---|---|
| **Delete V** | F | GCAATAAATCGTCCAATGCTGGTTGACTATCAGCTAGATC |
| | R | GATCTAGCTGATAGTCAACCAGCATTGGACGATTTATTGC |
| **Insert ΔV1** | F | GCAATAAATCGTCCAATGCTAAGAAGGGCCCCTGGTCTAG |
| **Insert ΔV2** | F | GCAATAAATCGTCCAATGCTAAGAAAGGCCCCTGGTCTAG |
| **Insert ΔV1/ ΔV2** | R | GATCTAGCTGATAGTCAACCTTACTAACCCTTTGCGAAATAG |
| *SDM F117S | F | AATAATGGCGCCTATAGAGCGCCTCTGTCTCCG |
| | R | GGAGACAGAGGCGCTCTATAGGCGCCATTATT |
| **SDM ICD** | F | CTAGCATGCGCCCTAATGTAC |
| | R | GTACATTAGGGCGCATGCTAG |
| **SDM TIS F** | F | CGCCCTCCAGGGCCACCATGGGCTCCAGACC |
| | R | GGTCTGGAGCCCATGGTGGCCCTGGAGGGCG |
| **SDM TIS HN** | F | CACCGACAACAGTCCGCCACCATGGACCGCGC |
| | R | GCGCGGTCCATGGTGGCGGACTGTTGTCGGTG |
| **Sequence analysis V** | F | GGGTAAACCAGCAGAG |
| | R | CTTGCTTAGGAGCTTGGC |
| **Sequence analysis MBCS** | F | GGCCAAGATACTCTGGAG |
| | R | GTAAAGTGCCTGAATAG |

*SDM: site directed mutagenesis

(Table 2) as described in Liu, 2008 [33]. Additional mutations were introduced by side-directed mutagenesis as described before using specific primers (Table 2) [8]. Subsequently, the different subclones were cloned back to the full-length construct and sequenced for their correctness using a 3130xL Genetic Analyzer (Life Technologies).

## Sequencing

RNA was extracted from the virus stocks using the High Pure RNA isolation kit (Roche Diagnostics, The Netherlands) according to manufacturer's instructions. cDNA was produced as described before using the Superscript III Reverse Transcriptase kit (Invitrogen, Thermo Fischer) [34]. Primers (Table 3) were used to amplify seven overlapping parts of the genome using Pfu Ultra II Fusion HS NA Polyemarase (600674, Agilent technologies, USA) followed by purification using the MinElute Purification Kit (Qiagen, Germany). Sequence primers were used to sequence the V gene and the MBCS of the F gene using a 3130xL Genetic Analyzer (Life Technologies) (Table 2).

## Rescue and passaging of recombinant viruses

Recombinant NDVs were rescued by transfecting BSR-T7 cells with 5 μg full length NDV plasmid, 2.5 μg pClneo-NP, 1.25 μg pCleo-P and 1.25 μg pClneo-L using 8 nM calcium phosphate. Three days later, BSRT-7 cells were scraped and co-cultured with Vero cells. After 5 days, co-cultures were harvested and used to generate passage 1 in Vero cells. Titrations were done in Vero cells as described before [8]. Passage 2 was produced in Vero cells using an MOI 0.01, titrated and stored at -80˚C. In case of NDV F0, BSRT-7 cells were scraped, and the cell suspension was then inoculated in to 10-day old eggs. Following passages were produced in Vero cells in the presence of 2 μg/ml TPCK-treated Trypsin (T1426, Sigma-Aldrich). For all experiments virus stocks from passage 2 were used unless indicated otherwise. The titer of the virus

**Table 3. Amplification primers.**

| Function | Amplicon size | Primer | Sequence 5'- 3' |
|---|---|---|---|
| Amplicon 1 | 2232 | F | ACCAAACAGAGAATCCGTGAGTTACG |
|  |  | R | GTTGCTTGCTCCGGTCCTGAG |
| Amplicon 2 | 2340 | F | CAGCATGGGAGAAGCATGGGAG |
|  |  | R | GGATTGTATTTGGCAAGGGTGTGCC |
| Amplicon 3 | 2172 | F | GCCAAGATACTCTGGAGTCAAACCG |
|  |  | R | GCTTCACCGACAACAGTCCTC |
| Amplicon 4 | 2404 | F | GTGTGAAAGTTCTGGTAGTCTGTCAG |
|  |  | R | GGAAGCGGTAGCCCAGTTAATTTCC |
| Amplicon 5 | 2375 | F | GTGGCAATGAGATACAAGGCAAAACAGC |
|  |  | R | GGCTTGATGCAACTGTGTCAACACC |
| Amplicon 6 | 2421 | F | GCCAGAAGCTATGGACAATGATCTC |
|  |  | R | CTGCAAGTTGGTGTGATCCGTCATG |
| Amplicon 7 | 3298 | F | GAAGTGCTCCTCGACTGTTCTTACC |
|  |  | R | ACCAAACAAAGATTTGGTGAATGACGAGAC |

stocks was determined by end-point titration in Vero cells, calculated using the method of Reed & Muench and expressed as $TCID_{50}$ $ml^{-1}$ [35].

## Replication kinetics

For all cell types, one million cells were seeded in 6-well plates (Corning). The next day, cells were inoculated at the indicated multiplicity of infections (MOIs) and after 1h incubation cells were washed three times with phosphate buffered saline (PBS) and cell-specific media was added. In case of IFN pre-treatment, cells were pretreated for 24h with 30 μg/ml chicken IFN-β (abx067344, Abbexa, UK). At the indicated time points, 100 μl supernatant was collected and stored with 25% sucrose (w/w) at -80˚C. Subsequently, collected samples were titrated in Vero cells. To this end, 24- well plates (Corning, The Netherlands) were seeded with 200.000 cells per well and inoculated with virus and supernatant was collected 48 hours after washing. All samples for the NDV F0 virus were titrated in media supplemented with 2 μg/ml TPCK-treated Trypsin (T1426, Sigma-Aldrich).

## Replication kinetics *in ovo*

Embryonated chicken and duck eggs of various ages were inoculated in the allantoic cavity with 100 μl 1E6 $TCID_{50}$/ml virus. After 48 hours incubation at 37˚C, the allantoic fluid was harvested and stored at -80˚C until titration in Vero cells as described before, using infection media without trypsin supplementation.

## Determination of virus induced IFN-β mRNA expression levels

DF-1 or A549 cells were seeded in 24-well plates (Corning) and inoculated at an MOI of 0.01 (DF-1) or 1 (A549). In case of A549 cells, spin-inoculation (800xg, 10 min) was applied [36]. After 1 hour, cells were washed once with PBS and fresh infection media was added. Subsequently, cells were incubated for 24 hours and RNA was isolated according to the manufacturer's instructions. In addition, cells were treated with trypsin-EDTA (Lonza) and fixed in Cytofix/Cytoperm (BD, The Netherlands) for flow cytometry analysis according the manufacturer's instructions. After fixation, cells were incubated in 1% normal goat serum (MP Biomedicals) and stained with 1:1000 anti-NDV (ab34402, Abcam, UK) and 1:1000 secondary FITC-

labelled antibody (ab6749, Abcam) and analyzed by FACS Canto (BD) to determine the percentage of infected cells. qRT-PCR (45 cycles) was performed using 5 µl RNA in an ABI PRISM 7500 sequence Detection System (Life Technologies) using TaqMan gene expression assay for human IFN-β, chicken IFN-β and chicken β-actin (Hs00277188, Gg03344129, Gg03815934, Thermo Fischer Scientific). Human β-actin primers have been described before [31]. Results are shown as fold change of inoculated samples versus mock-inoculated samples both corrected for the household gene actin-β, calculated using the $2^{-\Delta\Delta}{}_T$ method [37].

### Cytotoxicity assay

HPAC cell lines were seeded in 96-well plates (Greiner) and inoculated at the indicated MOIs. After 1 hour, cells were washed once with PBS and fresh infection media was added. A lactate dehydrogenase assay (Promega, The Netherlands) was used to determine cell viability as described before [31].

### Fusion assay

One million cells were seeded in 6-well plates (Corning) and inoculated at an MOI of 1. After 1 hour, cells were washed once with PBS and fresh infection media was added. Subsequently, cells were incubated for 16 hours and fixed with 4% PFA. The cells were then stained with Giemsa (HX71780604, Millipore) and the fusion index was determined by averaging the number of nuclei of 30 fusion foci.

### Mean death time assay

Ten-day-old embryonated specific pathogen free (SPF) chicken eggs were inoculated with 100 µl 1E6 TCID50/ml virus and incubated at 37˚C for up to 160 hours. All eggs were candled every 8 hours to determine whether the embryo was still alive.

### Supporting information

**S1 Data. Raw data used for Figs 1, 2, 3, 4 and 5 provided in separated tabs.**
(XLSX)

### Author Contributions

**Conceptualization:** Casper H. J. van Eijck, Ron A. M. Fouchier, Bernadette G. van den Hoogen.

**Data curation:** J. Fréderique de Graaf, Stefan van Nieuwkoop, Theo Bestebroer, Daphne Groeneveld.

**Formal analysis:** J. Fréderique de Graaf, Stefan van Nieuwkoop, Bernadette G. van den Hoogen.

**Funding acquisition:** Casper H. J. van Eijck, Ron A. M. Fouchier, Bernadette G. van den Hoogen.

**Investigation:** J. Fréderique de Graaf, Theo Bestebroer, Daphne Groeneveld.

**Methodology:** J. Fréderique de Graaf, Stefan van Nieuwkoop, Theo Bestebroer, Daphne Groeneveld.

**Project administration:** Bernadette G. van den Hoogen.

**Supervision:** Casper H. J. van Eijck, Ron A. M. Fouchier, Bernadette G. van den Hoogen.

**Writing – original draft:** J. Fréderique de Graaf, Bernadette G. van den Hoogen.

**Writing – review & editing:** J. Fréderique de Graaf, Ron A. M. Fouchier, Bernadette G. van den Hoogen.

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
