## [Decision Letter · Decision Letter 0]

7 Jan 2022

PONE-D-21-05320Optimizing environmental safety and cell-killing potential of oncolytic Newcastle Disease virus with modifications of the V, F and HN genesPLOS ONE

Dear Dr. van den Hoogen,

Thank you for submitting your manuscript to PLOS ONE. After careful consideration, we feel that it has merit but does not fully meet PLOS ONE’s publication criteria as it currently stands. Therefore, we invite you to submit a revised version of the manuscript that addresses the points raised during the review process.

Please integrate the suggestions brought forward by the reviewer into the manuscript. Where this might not be possible, please discuss the reasons. Furthermore, please expand the discussion part of the manuscript, which is currently still too short. Please add two paragraphs there:1. Please explain the limitations of your results in detail.2. Please discuss what advantages and disadvantages NDV might have in comparison to other oncoloytic viruses.

We look forward to receiving your revised manuscript.

Kind regards,

Michael C Burger, M.D.

Academic Editor

PLOS ONE

Journal Requirements:

"We would like to acknowledge the support from the Dutch Foundation OAK (“Overleven with alvleesklierkanker”) and NWO-TTW grant #15414 (NWO-domein Toegepaste en Technische Wetenschappen)."

"JFG and DG obtained funding from the Dutch Foundation “Overleven with alvleesklierkanker”; https://www.supportcasper.nl/nl/over-support-casper/stichting/

RF and SN obtained funding from  NWO-TTW grant #15414 (NWO-domein Toegepaste en Technische Wetenschappen;

 https://www.nwo.nl/toegepaste-en-technische-wetenschappen-ttw).

Reviewers' comments:

Reviewer's Responses to Questions

**Comments to the Author**

1. Is the manuscript technically sound, and do the data support the conclusions?

Reviewer #1: Yes

2. Has the statistical analysis been performed appropriately and rigorously? 

Reviewer #1: Yes

3. Have the authors made all data underlying the findings in their manuscript fully available?

Reviewer #1: Yes

4. Is the manuscript presented in an intelligible fashion and written in standard English?

Reviewer #1: Yes

5. Review Comments to the Author

Reviewer #1: The manuscript “Optimizing environmental safety and cell-killing potential of oncolytic Newcastle Disease virus with modifications of the V, F and HN genes” is a well-reported high-quality study. The goal of the study is to make the Newcastle disease virus (NDV) less harmful to poultry, while maintaining its oncolytic properties. Achieving this goal is difficult, and the study's authors along with some other researchers took up the challenge.

The authors of the study create NDV constructs that disrupt the translation of the viral protein V, which is responsible for blocking the interferon response in avian host cells. The disruption occurs by introducing one- or two-point mutations in the stutter site and introducing several stop codons into the gene encoding the V protein, without damaging the translation of another viral gene, which is partially translated from another coding frame. Next step, which is described in the manuscript, includes verification through multiple experiments, that this disruption indeed makes the viral constructs less infectious for avian cells and embryos. After this step, the authors through multiple virus passaging checked the genetic stability of mutants. By monitoring the frequency of reverse mutations, they found that the created mutants do not return to the original variant with an intact V gene, but after passaging in human cells they developed another frequent mutation, leading to one amino acid change in viral F-gene. However, this unexpected gain of function for F gene manifest itself in better viral fusogenic activity of human cells, but not in higher virulence for avian cells or embryos. Thus, the authors concluded that created viral constructs are attenuated and less virulent for avian species.

Unfortunately, screening in various cancers cell lines demonstrated that the constructs to same degree lost their oncolytic properties. Some cells line demonstrated less sensitivity to viral infection by created constructs. To solve this problem, further genetic engineering was carried out, which led to an improvement in the oncolytic properties of the constructs without gaining avian virulence. Consequently, there are more gains compared to losses.

While the best tradeoff between avian virulence and oncolytic virus properties is still to come for NDV, the authors of the manuscript have made valuable contributions to this area by better pinpointing the genes that are the main players in the game. Moreover, some developed NDV constructs are candidates for preclinical studies in animals.

In my opinion, the main claims of the article are supported by the evidence provided. The conclusions, together with the information that supports these conclusions, serve the research area of oncolytic viruses well. I think that the scientific context of the study and previous research are well presented. The experimental work is logical, each step is justified, and the results are well communicated.

However, I do suggest some improvements.

Major:

Discussion section

The authors use Vero cells to titrate variable viral constructs grown in different cancer cell lines. However, they show that in Vero cells, different constructs grew with variable efficiency (Fig. 4D). For example, there was a significant difference in growth curves for the NDV3aa and NDV 3aa stop V constructs. This difference in growth curves indicates that titration of Vero cells may cause some experimental error. Consequently, experimental measurements of the titers of viruses produced in different cell lines systematically introduce some bias. Perhaps the weak correlation between the efficacy of malignant cells killing efficiency by the viral constructs and their titer measured in Vero can to some extent can be explained by this introduced bias. I think it would be nice if the authors paid attention to this issue in the discussion section.

Minor:

Annotation

The text is duplicated in the annotation on the first page of the pdf file.

Introduction

The Introduction section can be improved by adding one or more sentences to indicate the mechanism by which the V gene acts as an alpha interferon antagonist in avian cells and why the mechanism appears to be species specific. Also, other recent studies in relation to oncolytic NDV can be cited such as: “Genetic Modification of Oncolytic Newcastle Disease Virus for Cancer Therapy” https://www.ncbi.nlm.nih.gov/labs/pmc/articles/PMC4934751/ and

“Oncolytic Newcastle disease virus activation of the innate immune response and priming of antitumor adaptive responses in vitro” https://www.ncbi.nlm.nih.gov/labs/pmc/articles/PMC7230062/

Figures

1) Figures would benefit if more information from the legends were added directly to the images. Thus, it will be easier to understand the graphical information if the cell lines, their origin (human or avian) and the names of the viral constructs are indicated directly in the figures, and not in their text legends.

2) Line 471. I think that authors meant “Results are represented as fold change of IFN mRNA transcription in mock versus virus-infected cells versus mock cells.” Otherwise, the sentence is meaningless along with the relevant histogram data, it looks like the mock infected A549 cells produce more interferon compared to the infected ones.

3) What is an explanation for Figure 2 B results? Why NDV F3aa- STOPV1, is less efficiently inhibits interferon mRNA compared to NDV F3aa- STOPV2 in avian DF-1 cells?

References

There are two weird references 10 and 11 (line 378).

Some other minor suggestions and corrections in the attached file.

6. PLOS authors have the option to publish the peer review history of their article (what does this mean?). If published, this will include your full peer review and any attached files.

Reviewer #1: **Yes: **Olga Matveeva

---

## [Author Response · Author response to Decision Letter 0]

24 Jan 2022

Comments from the Reviewer:

Annotation

The text is duplicated in the annotation on the first page of the pdf file.

We thank the reviewer for pointing out. We have tried to correct this with the re-submission, but, unfortunately, in the process of building the PDF this happened again. I hope the editor knows how to avoid this.

Introduction

The Introduction section can be improved by adding one or more sentences to indicate the mechanism by which the V gene acts as an alpha interferon antagonist in avian cells and why the mechanism appears to be species specific. Also, other recent studies in relation to oncolytic NDV can be cited such as: “Genetic Modification of Oncolytic Newcastle Disease Virus for Cancer Therapy” https://www.ncbi.nlm.nih.gov/labs/pmc/articles/PMC4934751/ and

“Oncolytic Newcastle disease virus activation of the innate immune response and priming of antitumor adaptive responses in vitro ”https://www.ncbi.nlm.nih.gov/labs/pmc/articles/PMC7230062/

We thank the reviewer for this comment, and we have now added text on the function of V and added the references.

Line 46: as the V protein efficiently prevents innate host defences, such the type I IFN response as well as apoptosis, in avian cells and not human cells.

Line 41: Several studies aimed to lower the virulence of mesogenic NDV-73T strain, while keeping the oncolytic efficacy of the virus, either by changing the cleavage site to that of a non-virulent virus, or by insertion of 198 nt at the HN-L junction. Alternatively, NDV-73T was armed with immune modulatory protein genes such as granulocyte–macrophage colony-stimulating factor (GM-CSF), to increase the oncolytic effect of the virus (REF).

Discussion section

The authors use Vero cells to titrate variable viral constructs grown in different cancer cell lines. However, they show that in Vero cells, different constructs grew with variable efficiency (Fig. 4D). This difference in growth curves indicates that titration of Vero cells may cause some experimental error. Consequently, experimental measurements of the titers of viruses produced in different cell lines systematically introduce some bias. Perhaps the weak correlation between the efficacy of malignant cells killing efficiency by the viral constructs and their titer measured in Vero can to some extent can be explained by this introduced bias. I think it would be nice if the authors paid attention to this issue in the discussion section.

We thank the reviewer for this comment. 

We thank the reviewers for suggesting possible explanations for the weak correlation between killing efficiency and measured titer in Vero cells.

We like to point out that Vero cells are the most susceptible cells for NDV and its variants and is therefore the first choice for titrations. In addition, the method used for titration is an end-point dilution assay, not a replication assay. This means that the input is serial diluted, in theory to 1 virus particle per well, and 5 days after inoculation the wells are scored for positive or negative, allowing sufficient time to detect the presence of virus. As this scoring is based on positive/negative, and not on replication, attenuation in Vero cells would not influence the outcome of the titrations. 

The possible bias that the reviewer refers to is not always that clearly reflected in Fig 3a and b. While NDV F0 and the NDV F3aa- STOPV viruses reached equal titers in CFPAC, inoculation with F0 induced more cell death that inoculation with STOPV. If titers for the STOPV were underestimated, and if there was a correlation between cell killing efficiency and titers, you would expect higher killing efficiency of STOPV compared to F0 in these cells. 

Figures

1) Figures would benefit if more information from the legends were added directly to the images. Thus, it will be easier to understand the graphical information if the cell lines, their origin (human or avian) and the names of the viral constructs are indicated directly in the figures, and not in their text legends.

We thank the reviewer for the suggestion, and we have changed the figures accordingly.

2) Line 471. I think that authors meant “Results are represented as fold change of IFN mRNA transcription in mock versus virus-infected cells versus mock cells.” Otherwise, the sentence is meaningless along with the relevant histogram data, it looks like the mock infected A549 cells produce more interferon compared to the infected ones.

Line 472: the sentence has changed to: “mRNA transcription in virus-infected cells versus mock”

3) What is an explanation for Figure 2 B results? Why NDV F3aa- STOPV1, is less efficiently inhibits interferon mRNA compared to NDV F3aa- STOPV2 in avian DF-1 cells?

We think the reviewer means ‘why does stopV1 induces more interferon production than stopV2’, because viruses without expression of V are not able to inhibit interferon production. 

The difference in induction of interferon production between NDV F3aa-STOPV1 and STOPV2 is likely due to the higher level of attenuation of NDV F3aa-STOPV2 in avian cells. This is shown in the replication kinetics in figure 1C, where at 24 hrs after infection the titers for NDV F3aa-STOPV2 are lower than those of STOPV1. More pronounced is the effect in cells that are pre-treated with interferon (fig2A) or in older chicken eggs, producing interferon, as shown in fig 2D.

We added the following lines (line 103): “Inoculation of DF-1 cells with NDV F3aa-STOPV2 resulted in lower IFN IFN-β mRNA expression levels than inoculation with NDV F3aa-STOPV1, which is probably due to the higher levels of attenuation of NDV F3aa-STOPV2 in cells pretreated or producing IFN, as shown in figure 2A”.

And at line 116: “underpinning the higher degree of attenuation of NDV F3aa-STOPV2 compared to that of NDV F3aa-STOPV1.”

References

There are two weird references 10 and 11.

References 10 and 11 have now been corrected

Some other minor suggestions and corrections in the attached file.

We made the changes as recommended by the reviewer.

Line 69: “The paragraph is too long. The reviewer suggests to start a new paragraph from the sentence “Recombinant viruses..” (which is now line 77 in revised version)

We have started a new paragraph. However, not with the sentence the reviewer referred to, as we think this sentence belongs with the previous paragraph. We did start a new paragraph at line 81, with a new heading for the first paragraph ‘Generation of NDV F3aa lacking V protein expression’

Line 177: we have added the words ‘the viruses’ (line 194 in revised manuscript)

Line 215: The paragraph is too long. The text will be easier to understand if it is split into two or more smaller paragraphs.

The paragraph (now starting at line 249 in the revised manuscript) has now been split up in two paragraphs (break at line 257). 

Line 470 (line 514 in revised version) : the sentence has changed to: “mRNA transcription in virus-infected cells versus mock”

Comments from the editor:

Please expand the discussion part of the manuscript, which is currently still too short. Please add two paragraphs there:

1. Please explain the limitations of your results in detail.

2. Please discuss what advantages and disadvantages NDV might have in comparison to other oncoloytic viruses.

We have now added a paragraph on the limitations of our results and a paragraph on the advantages and disadvantages of NDV as oncolytic virus,.

1. Line 270: “However, it is not known yet whether the virus is able to cause disease in adult chickens. This is important information for the unlikely event that treated patients spread virus to birds. In addition, at present, all NDV strains which possess an MBCS in their fusion protein, or have an ICPI above 0.7, are considered to be select agents. Like the virulence in adult chickens, the ICPI has not been determined yet for the NDV F3aa-STOPV viruses. Based on the MDT, it is expected that these viruses have an ICPI lower than 0.7, classifying the viruses as lentogenic. However, the presence of an MBCS in the fusion protein, would still categorize the virus as virulent. Perhaps, an attenuated strain of a virus with an MBCS could be approved for viro immune-therapy studies by the regulatory instances when virulence studies indicate the attenuated nature of these viruses”.

2. Line 212: “Oncolytic viro immunotherapy, based on OVs, is rapidly gaining interest in the field of immunotherapy against cancer. Due to significant virulence associated with the use of some of the human pathogens, animal viruses have been explored as an alternative, with NDV being one of them. NDV does not cause disease in humans, the virus does not interact with the host cell DNA, cellular genome integration is impossible, and the virus can efficiently and selectively replicate in tumor cells. In addition, NDV has been shown to be very safe upon high dose administration in humans, with no serious adverse events noted in early clinical trials. The results of early trials with NDV have been relatively disappointing, but with the advent of recombinant DNA techniques it has become possible to genetically engineer NDV and interest in the use of recombinant NDV (rNDV) as an oncolytic virus has revived over the last decade. One of the advantages of NDV is that genetic engineering of the genome is relatively easily, and the genome is forgiving for the acceptance of foreign genes. However, the limitation of using NDV is their potential to become virulent avian species”. 

Please ensure that your manuscript meets PLOS ONE’s style requirements.

We have made changes accordingly.

We note that the grant information you provided in the ‘Funding Information” and “Financial Disclosure” sections do not match. When you resubmit, please ensure that you provide the correct grant numbers for the awards you received for your study in the ‘Funding Information’ section.

We have corrected this.

We note that you have provided funding information that is not currently declared in your Funding Statement. However, funding information should not appear in the Acknowledgments section or other areas of your manuscript. Please remove any funding-related text from the manuscript and let us know how you would like to update your Funding Statement. Currently, your Funding Statement reads as follows: "JFG and DG obtained funding from the Dutch Foundation “Overleven with alvleesklierkanker”; https://www.supportcasper.nl/nl/over-support-casper/stichting/

RF and SN obtained funding from NWO-TTW grant #15414 (NWO-domein Toegepaste en Technische Wetenschappen;

https://www.nwo.nl/toegepaste-en-technische-wetenschappen-ttw).

We have taken out the information on funding in the acknowledgement section. And we have removed the acknowledgement section.

In your Data Availability statement, you have not specified where the minimal data set underlying the results described in your manuscript can be found. PLOS defines a study's minimal data set as the underlying data used to reach the conclusions drawn in the manuscript and any additional data required to replicate the reported study findings in their entirety. All PLOS journals require that the minimal data set be made fully available. For more information about our data policy, please see http://journals.plos.org/plosone/s/data-availability.

We have now made the minimal data set available.

We note that you have included the phrase “data not shown” in your manuscript. Unfortunately, this does not meet our data sharing requirements. PLOS does not permit references to inaccessible data. We require that authors provide all relevant data within the paper. If the data are not a core part of the research being presented in your study, we ask that you remove the phrase that refers to these data.

The data is not a core part of the research. The described results are in fact depicted in fig 1B. We have now indicated this in the text. Line 79.

We have reviewed our reference list. We have not cited papers that have been retracted. We have improved reference 10 and 11.

---

## [Editor Report · Decision Letter 1]

26 Jan 2022

Optimizing environmental safety and cell-killing potential of oncolytic Newcastle Disease virus with modifications of the V, F and HN genes

PONE-D-21-05320R1

Dear Dr. van den Hoogen,

We’re pleased to inform you that your manuscript has been judged scientifically suitable for publication and will be formally accepted for publication once it meets all outstanding technical requirements.

Kind regards,

Michael C Burger, M.D.

Academic Editor

PLOS ONE
---

## [Editor Report · Acceptance letter]

31 Jan 2022

PONE-D-21-05320R1 

Optimizing environmental safety and cell-killing potential of oncolytic Newcastle Disease virus with modifications of the V, F and HN genes 

Dear Dr. van den Hoogen:

I'm pleased to inform you that your manuscript has been deemed suitable for publication in PLOS ONE. Congratulations! Your manuscript is now with our production department. 

Kind regards, 

on behalf of

Dr. Michael C Burger 

Academic Editor

PLOS ONE